# ASRS Questionnaire and Tobacco Use: Not Just a Cigarette. A Screening Study in an Italian Young Adult Sample

**DOI:** 10.3390/ijerph18062920

**Published:** 2021-03-12

**Authors:** Lorenzo Zamboni, Pierpaolo Marchetti, Alessio Congiu, Rosaria Giordano, Francesca Fusina, Silvia Carli, Francesco Centoni, Giuseppe Verlato, Fabio Lugoboni

**Affiliations:** 1Department of Medicine, Addiction Medicine Unit, Verona University Hospital, 37134 Verona, Italy; alessio.congiu@hotmail.it (A.C.); rosariagiordanovr@gmail.com (R.G.); sil.carli88@gmail.com (S.C.); francescocentoni1994@hotmail.it (F.C.); fabio.lugoboni@aovr.veneto.it (F.L.); 2Department of Neurosciences, Biomedicine, and Movement Sciences, University of Verona, 37134 Verona, Italy; 3Diagnostics and Public Health-Unit of Epidemiology and Medical Statistics, University of Verona, 37134 Verona, Italy; pierpaolo.marchetti@univr.it (P.M.); giuseppe.verlato@univr.it (G.V.); 4Department of General Psychology, University of Padova, 35131 Padova, Italy; francescafusina@gmail.com; 5Padova Neuroscience Center, University of Padova, 35131 Padova, Italy

**Keywords:** ADHD symptomatology, tobacco, benzodiazepine

## Abstract

Young adults exhibit greater sensitivity than adults to nicotine reinforcement, and Attention Deficit Hyperactivity Disorder (ADHD) increases the risk for early-onset smoking. We investigated the correlation between ADHD Self-Report Scale (ASRS) scores and smoking, evaluated the prevalence of ADHD symptomatology (not diagnoses) in smokers and non-smokers and its comorbidity with benzodiazepine and gambling addictions. A total of 389 young adults from 14 schools in Northern Italy fill out a survey and the Adult ADHD Self-Report Scale (ASRS). A total of 15.2% of subjects tested positive at the ASRS, which correlated with smoking; moreover, smokers had twice the probability of testing positive at the ASRS. ADHD symptomatology, especially when comorbid with tobacco abuse, is an important condition to monitor because early nicotine exposure could be a gateway for other addictive behaviors.

## 1. Introduction

Attention Deficit Hyperactivity Disorder (ADHD) is a neuropsychiatric disease that literature indicates to be among the most prevalent disorders in children [1]. Studies estimate that it is diagnosed in 5% to 7% of children (aged 2–17) and in 4% of adults [2,3].

In these last few years, the conceptualization of the development of ADHD has changed. While this neuropsychiatric disorder was once thought to disappear with adolescence, long-term controlled follow-up studies have shown otherwise, indicating that the persistence of the disorder continues into adolescence in three quarters of cases and into adulthood for 50% of cases [4,5]. Specifically, some studies seem to indicate that hyperactivity, which manifests in childhood, usually tends to decrease in adolescents and young adults, being replaced by a sense of internal or mental restlessness [6]. Another study is in agreement with these data, showing how the symptoms of inattention tend to remain constant even in late adolescence, in contrast to the symptoms of hyperactivity, which decrease with age [7].

The discovery of the persistence of ADHD in adolescence and adulthood has progressively oriented the interest of researchers to study its implications in the development of different forms of psychological distress. Today we know that comorbidity between ADHD and other disorders is particularly frequent, especially for internalizing conditions (13–15%), such as anxiety and depressive disorders, and externalizing disorders (43–93%), such as oppositional-defiant and conduct disorders [8]. Among these, anxious–depressive symptoms seem to be the most frequent internalizing problems [9], while Substance Use Disorder (SUD) is the most common problem among externalizing ones.

SUD is a psychiatric disorder with implications in healthcare, lost workplace productivity, crime and other social problems [10]. Approximately 9% of young adults manifest a SUD and 6% could be included in an alcohol use disorder classification [11]. The severity of the SUDs, the decreased efforts to seek treatment and the prolonged duration of SUDs in adulthood are predicted by a premature onset (childhood) [12,13]. Early-onset SUDs are associated with elevated rates of risky behaviors: suicidal tendencies, academic failure, employment problems and the like, are among the most relevant examples [14,15].

Several studies recognize ADHD as one of the predisposing factors for the development of SUDs. In a recent meta-analysis, for example, van Emmerik and associates [16] showed that 23% of treatment-seeking young adult substance abusers had ADHD. In agreement with this result are McAweeney and colleagues [17], who found a significant difference in the prevalence rate of SUDs in youth previously diagnosed with ADHD (3%) and those diagnosed while in treatment (44%).

A high rate of ADHD symptoms was found among heroin-dependent patients, particularly those affected by the most severe forms of addiction. These individuals had higher rates of unemployment, other co-morbid mental health conditions, and were heavy tobacco smokers [18].

Among the various substances that are abused, tobacco is the one most associated with ADHD [19]. The tobacco epidemic is a major factor that threatens human health and economic development. With one billion smokers in the world and six million deaths per year [20], nicotine addiction is the most frequent SUD in the world. Of all, young adults may represent the population which is most at risk of developing this specific form of addiction. Several studies indicate how they are more sensitive to the reinforcing effects of nicotine, which increases the susceptibility of young adults to nicotine dependence when comparing them to their adult counterparts [21]. Their reduced smoking history makes young adults relatively lighter smokers, although the presence of withdrawal and negative reinforcement processes associated with smoking re-establishment are already involved in this initial phase of nicotine addiction [22,23].

It is well described that ADHD symptomatology increases the risk for earlier tobacco smoking initiation, from occasional to regular smoking, to developing more severe nicotine addiction and having more failed quit attempts [24,25,26,27,28,29]. A higher prevalence of co-morbid psychiatric symptoms and co-occurring heavy tobacco addiction was found in heroin-dependent patients with a positive screening test for ADHD symptoms compared to patients who did not screen positive for ADHD symptoms. Although most of the patients with drug use disorders were also tobacco smokers, heavy smokers were significantly more likely to meet ADHD symptom criteria than non-smokers [18].

Nicotine withdrawal severity is a predictor of smoking maintenance and relapse. In ADHD smokers, this process appears augmented [30]. Several papers reported more withdrawal severity, greater irritability and difficulty concentrating during smoking abstinence [24,31,32,33]. Kollins and colleagues [32] suggest that the withdrawal severity and the desire to relieve it (especially after a period of abstinence) heighten the reinforcing effects of smoking in ADHD smokers more than in non-ADHD smokers.

The neurobiological bases of ADHD have often been called into question in explaining the predisposition of this increased sensitivity to nicotine that was found in several studies. This particular process could be explained with an alteration in dopamine and other catecholaminergic functions in the brain regions and neuro-transmitter systems that are also modulated by both chronic and acute nicotine administration [34,35]. The increasing risk of smoking progression is therefore linked with the neurobiological bases of ADHD, conferring heightened vulnerability to the reinforcing effects of nicotine, especially during withdrawal [32,36].

The predisposition to use cigarettes found in young adult populations with ADHD acquired relevance not only for the adverse effects related to smoking; several studies indicate that the use of cigarettes during early adolescence often precedes the use of illicit substances during late adolescence [37], representing a further predisposing factor for the development of SUDs. Therefore, if the presence of ADHD in adolescence facilitates the subsequent use of substances [38], the simultaneous presence of smoking could further increase this risk, and indeed this is what has been found in recent studies [39,40,41,42].

For these reasons, our study has three main objectives: (1) to observe the prevalence and correlation between ADHD and tobacco use; (2) to evaluate the risk of positive ADHD screening in young adult smokers and non-smokers; (3) to evaluate other possible addiction comorbidities, such as benzodiazepine (BDZ) addiction, alcohol addiction and gambling disorder (GD).

We expect a higher prevalence of smokers with a positive ADHD screening questionnaire and a higher odds ratio in the same class, in line with the evidence currently present in literature [5,43], as well as a higher prevalence of alcohol, GD and BDZ use/misuse in young adult smokers compared with non-smokers.

## 2. Materials and Methods

The sample comprised 389 persons aged between 18 and 22 years old who were enrolled in 14 schools in Northern Italy; of these, 9 schools were located in Mantua and 5 in Verona.

Approval for the research was obtained from the Human Research Approval Committee (CARU) of the University of Verona (approval code: 9/2019).

An anonymous and self-administered questionnaire composed by ad hoc questions was used. It was composed of 17 multiple-choice questions and 6 open questions, for a total of 23 questions.

The survey was presented and distributed in classrooms by psychologists from January to June 2019. The time needed to complete it was about 15 min; participants filled out the survey by themselves using paper and pencil. Participation in the survey was voluntary and without remuneration; each young adult could have suspended the compilation at any time without providing any explanation.

The survey included questions regarding socio-demographical data, alcohol consumption, the use of benzodiazepines (BDZ) and gambling behavior. The survey areas were: sex, family relationships, licit and illicit drug use (alcohol, BDZ, THC, etc.), and gambling behavior. Moreover, the survey included the Adult ADHD Self-Report Scale (ASRS-v1.1) Symptom Checklist Part A [44], which comprises the six most predictive of the eighteen DSM-IV-TR criteria [10]. With a cut-off score ≥ 4, the six-question ASRS-v1.1 was found to outperform the full 18 ADHD criteria checklist with good sensitivity (68.7%) and *k* (0.76) and very high specificity (99.5%) and total classification accuracy (97.9%) [44]. The ASRS-v1.1 Symptom Checklist Part A was found to be a sensitive screener for identifying possible ADHD cases with very few missed cases among those screening negative in a large SUD population [45].

Descriptive statistics were performed using percentages for categorical variables and a mean and standard deviation for quantitative variables. Pearson’s chi-square test was used to evaluate the association between demographic characteristics, habits and adult ADHD symptoms in bivariate analyses. Multivariable logistic regression models were performed to assess the association between adult ADHD symptoms (<4 points vs. ≥4 points) and subjects’ habits such as smoking and benzodiazepine use. Specifically, we evaluated smoking as: daily smokers and smokers for at least one year (yes/no), number of daily cigarettes (none/1–10/>10), THC smokers in the last 12 months (yes/no) and benzodiazepine users (no, one time/more than one time). As covariates, we included in the model: sex, age, parents separated/divorced (yes/no), number of working family members (<2/2/>2), foreign father (yes/no), school (high school/technical institute/professional institute), other habits such as alcohol unit consumption (none/≤2/>2) and gambling (none, one time/more than one time). These associations were evaluated using odds ratios (OR) and 95% confidence intervals (CI). Statistical analyses were performed with STATA 16 (Stata Corp. College Station, TX, USA) and statistical significance was set at *p*-value < 0.05.

## 3. Results

Participants in this study were 389 young adults, 63.5% of which were male. The mean age in the sample was 18.4 ± 0.6 years (range 18–22 years). Daily smokers were 105 (27%) and 80% of them smoked between 1 and 10 cigarettes every day. Young adults who smoked for at least one year were 92 (23.6%), and 119 (30.6%) used THC in the preceding 12 months. Of the subjects, 349 (89.7%) drank alcohol and 210 (54.4%) consumed two or more alcoholic units in the preceding year. Young adults who used BDZ in a lifetime period were 28 (7.2%) and 4.1% used BDZs more than one time. This study also showed that 112 (28.9%) participants presented a gambling behavior for two or more times in the last year (Table 1).

In the ASRS questionnaire, 59 (15.2%) subjects had positive results (≥4). No associations in the bivariate analyses were found about the demographic characteristics of the subjects, although 32.2% of those scoring positive on the questionnaire had separated or divorced parents, vs. 21.2% in those scoring negative, *p* = 0.064 (Table 2).

Table 3 shows the associations between the subjects’ habits and the ASRS questionnaire scores. The associations were statistically significant in smokers; a higher proportion of young adults who smoked were in the group scoring positive on the ASRS questionnaire. No associations were found about THC or BDZ use.

Multilevel regressions (Table 4) confirmed the associations found in bivariate analyses about smoking. Specifically, young adult smokers who smoked 1–10 cigarettes/day had more than twice the likelihood to score positive on the questionnaire in the model adjusted by all covariates (OR = 2.20, CI95%: 1.10–4.42). A similar result was found in young adults who smoked for more than one year (OR = 2.51, CI95%: 1.22–5.16), while no associations were found in young adults who used THC or BDZ.

## 4. Discussion

ADHD is one of the most prevalent neuropsychiatric disorders [1], but it is not frequently evaluated in young adults with SUDs or problematic drug use.

Several studies show that ADHD increases the likelihood of illicit drug use [39] and of tobacco use compared to peers [2,46,47]. Overall, it appears that people with ADHD symptomatology start smoking earlier, become more dependent [29] and have a greater risk of having continued cigarette use [28]. In the young adult population, the presence of ADHD symptoms facilitates early cigarette use [48] and the progressive increase in cigarettes compared to the absence of ADHD symptomatology [49].

Our study shows the correlation between ADHD symptomatology and tobacco use in young adult populations that also emerged in several other studies [19,50,51]. Specifically, young adult smokers present twice the possibility to score positive on an ADHD screening questionnaire (OR = 1.82, IC95% = 1.00–3.30) than non-smoking young adults. Compared to other studies [52,53,54], no similar results have been found in BDZ users, gambling and alcohol use. It is important to underline that we have used a screening questionnaire for ADHD, and not a diagnostic one.

In the current state of empirical research, the reason for which such a correlation is present has not yet been clarified. Among the hypotheses that have currently received greater support is one that views the use of cigarettes as a form of self-medication in the presence of ADHD symptoms [55,56], especially those related to distractibility [56] due to the effects that nicotine exerts in increasing performance and focus [57,58]. A possible objection to this hypothesis calls into question the self-medicating use of smoking that young adults with ADHD symptoms enact to reduce specific internalizing symptoms, such as anxious–depressive ones, which are frequent in this population. However, various pieces of evidence seem to invalidate this thesis: a recent study found that ADHD symptoms in young adults predicted alcohol abuse and cigarette consumption regardless of the presence of these internalizing symptoms [59]; another study also found that ADHD symptoms in adolescence and early adulthood predicted substance use regardless of the presence of other psychiatric conditions or behavioral problems [60]; an additional study, which was conducted with Mendelian randomization methods, found genetic polymorphisms in people struggling with SUDs and anxious and depressive symptoms. This evidence could suggest that the use of cigarettes is probably more associated with a self-medicating function, rather than being a risk factor for the emergence of this symptomatology, per se [61]. Consistent with this hypothesis is what emerged from Lee and colleagues’ study [62], which shows how the symptoms of ADHD in childhood increase the likelihood of early cigarette consumption in adolescence, which, in turn, increases the likelihood of subsequent illicit substance use.

Although there is currently a lack of studies that can explain the predisposing effects of ADHD symptomatology in the development of SUDs [41], the possibility that ADHD predisposes the development of SUDs by facilitating tobacco use has been receiving wide empirical support over the past few years.

The nicotine gateway theory is presented by several studies that point out how nicotine exposure in early adolescence in various rodent models increases: (a) the acquisition and intake of nicotine, alcohol, cocaine and methamphetamine; (b) the co-use of nicotine and alcohol; (c) the rewarding effects of nicotine, cocaine, methamphetamine and opioids [63].

It is important to explain that drugs of abuse (like marijuana, cocaine, alcohol and opioids) influence different neurotransmitter systems. They all exert their reinforcing effects via the mesolimbic system, which is a dopaminergic pathway that connects the Ventral Tegmental Area (VTA) to the Nucleus Accumbens (NAcc) [61,62].

This pathway’s protection, functions and development are influenced by GABA, glutamate, serotonin and acetylcholine [64,65]. Motivation and desire for rewarding stimuli and reward prediction are regulated by dopamine released into the NAcc [65,66]. As nAChRs modulate dopamine release, the gateway hypothesis posits that young adults’ nicotine exposure primes the brain’s reward system to enhance the reinforcing effects of drugs of abuse [57,67]. A study by Potter and Newhouse [68] shows that non-smoking young adults with ADHD combined type present improvements in cognitive performance following nicotine administration in several domains that are central to ADHD. Results from this study support the hypothesis that cholinergic system activity may be important in the cognitive deficits of ADHD and may be a useful therapeutic target.

Our study observed an off-label consumption of BDZs in the young adult population. Treatment of ADHD symptoms is therefore crucial in parallel with the treatment of drug addiction. The use of sleep medications is common in patients with ADHD starting from their childhood; up to 22% of children suffering from ADHD use hypnotic drugs, with no clear evidence for the safety and benefits of long-term use [69].

In our study, there was no significant correlation between BDZ consumption and ADHD symptoms. The reason for this lack of significance could reside in the insufficiency of the sample. It is, however, reasonable to suppose that nicotine use could expose young adults to a major risk of developing a BDZ addiction or problematic use, due to the influence that BDZs exert on the level of GABAergic transmission. Finally, our study explained that there is no correlation between smoking and GD. Future research could examine this aspect of considering environmental data besides the absence or presence of gambling behavior.

This study presents several limitations; the first one is the fact that using only a screening test of ADHD (ASRS) makes it impossible to clearly diagnose this pathology. Future studies should include the adult Connors self-report screening scale (CAARS-SV), which allows to determine the type of ADHD with a quantitative estimation of Inattentive symptoms, Hyperactive/impulsive symptoms and a total ADHD index [70]. We underline that an ADHD diagnosis can only be done through a proper clinical evaluation. It is crucial to keep in mind that no questionnaire should be confused with a precise psychiatric diagnosis. Other limitations of our study reside in the fact that we used a self-report questionnaire, and in the fact that the BZD sample we used was too small.

## 5. Conclusions

This paper confirms the relationship between tobacco abuse and ADHD symptomatology that was observed in other studies. Nicotine addiction is often considered a “minor” addiction problem by specialists (neuropsychiatry, psychiatry and addiction unit), but this substance can create strong links with other substances (cocaine, alcohol, cannabis and opioids). BDZ misuse and addiction in adolescents is not debated in the scientific community, but this phenomenon is making its way in this specific population. In our study, it was impossible to establish the relationship between screening ADHD symptomatology, nicotine and BDZ use, and neither to assess nicotine addiction severity, but future studies could explore this specific area using a bigger sample size.

## Figures and Tables

**Table 1 ijerph-18-02920-t001:** Frequency distribution of the sample.

Variables	Categories	*n*	%
**Sex**			
	Males	247	63.5
	Females	142	36.5
**Parents separated/divorced**			
	No	300	77.1
	Yes	89	22.9
**Number of working family members**			
	<2	86	22.2
	2	218	56.2
	>2	84	21.6
**Foreign father**			
	No	337	87.8
	Yes	47	12.2
**School**			
	High school	133	34.2
	Technical institute	197	50.6
	Professional institute	59	15.2
**Smoking**			
	No	284	73.0
	Yes	105	27.0
**Number of cigarettes/day**			
	Non-smokers	284	73.0
	1–10	85	21.9
	>10	20	5.1
**Smoking for at least 1 year**			
	No	297	76.4
	Yes	92	23.6
**Number of cigarettes/day smoked in the last year ***			
	Non-smokers	284	75.5
	1–10	73	19.4
	>10	19	5.1
**THC ** use in the last 12 months**			
	No	270	69.4
	Yes	119	30.6
**Alcohol (units)**			
	Non-drinkers	41	10.6
	1–2	135	35.0
	>2	210	54.4
**Benzodiazepine use**			
	None/one time	370	95.9
	Two or more times	16	4.1
**Gambling**			
	None/one time	275	71.1
	Two or more times	112	28.9

* Thirteen subjects were excluded because they were smokers for less than one year. ** Cannabinoid.

**Table 2 ijerph-18-02920-t002:** Characteristics of the subjects divided by positive ADHD Self-Report Scale (ASRS) questionnaire scores.

Variables	ASRS < 4 (*n* = 330) (%)	ASRS ≥ 4 (*n* = 59) (%)	*p*
**Sex**			
Males	63.6	62.7	0.892
Females	36.4	37.3	
**Age (Mean ± SD)**	18.34 ± 0.63	18.47 ± 0.70	0.216
**Parents separated/divorced**			
No	78.8	67.8	0.064
Yes	21.2	32.2	
**Number of working family members**			
<2	23.3	15.5	0.370
2	55.8	58.6	
>2	20.9	25.9	
**Foreign father**			
No	88.6	83.1	0.230
Yes	11.4	16.9	
**School**			
High school	33.6	37.3	0.706
Technical institute	50.6	50.8	
Professional institute	15.8	11.9	

**Table 3 ijerph-18-02920-t003:** Habits of subjects divided by positive ASRS questionnaire scores.

Variables	ASRS < 4 (*n* = 330) (%)	ASRS ≥ 4 (*n* = 59) (%)	*p*
**Smoking**			
No	75.2	61.0	0.024
Yes	24.8	39.0	
**Number of cigarettes/day**			
Non-smokers	75.2	61.0	0.050
1–10	19.7	33.9	
>10	5.1	5.1	
**Smoking for at least 1 year**			
No	78.8	62.7	0.007
Yes	21.2	37.3	
**Number of cigarettes/day smoked in the last year ***			
Non-smokers	78.0	62.1	0.019
1–10	17.0	32.8	
>10	5.0	5.2	
**THC use in the last 12 months**			
No	69.1	71.2	0.748
Yes	30.9	28.8	
**Benzodiazepine use**			
None/one time	96.0	94.9	0.694
Two or more times	4.0	5.1	

* Thirteen subjects were excluded because they were smokers less than one year.

**Table 4 ijerph-18-02920-t004:** Associations between subject habits and positive ASRS questionnaire evaluated by logistic regression models.

Variables	OR (95% CI) ^a^	OR (95% CI) ^b^	OR (95% CI) ^c^
**Smoking (cigarette/day)**			
Non-smokers	Ref.	Ref.	Ref.
1–10	2.00 (1.08–3.73)	2.09 (1.10–3.95)	2.20 (1.10–4.42)
>10	1.07 (0.29–4.03)	0.98 (0.25–3.84)	1.05 (0.26–4.19)
**Smoking for at least 1 year (cigarette/day)**			
No	Ref.	Ref.	Ref.
Yes	2.09 (1.14–3.84)	2.06 (1.10–3.86)	2.18 (1.11–4.30)
**Number of cigarettes/day smoked in the last year**			
Non-smokers	Ref.	Ref.	Ref.
1–10	2.28 (1.20–4.32)	2.32 (1.20–4.49)	2.51 (1.22–5.16)
>10	1.11 (0.29–4.21)	1.03 (0.26–4.05)	1.07 (0.26–4.33)
**THC use in the last 12 months**			
No	Ref.	Ref.	Ref.
Yes	0.92 (0.50–1.70)	0.88 (0.47–1.66)	0.82 (0.41–1.62)
**Benzodiazepine use**			
None/one time	Ref.	Ref.	Ref.
Two or more times	1.27 (0.35–4.61)	1.13 (0.29–4.37)	1.24 (0.32–4.87)

^a^, adjusted by sex and age; ^b^, adjusted by sex, age, parents separated/divorced, number of working family members, foreign father, school; ^c^, adjusted by sex, age, parents separated/divorced, number of working family members, foreign father, school, other habits (alcohol unit consumption and gambling); OR, odd ratio; CI, confidence interval.

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
