# Peer review of "ASRS Questionnaire and Tobacco Use: Not Just a Cigarette. A Screening Study in an Italian Young Adult Sample"

_ijerph, 2021, doi:10.3390/ijerph18062920_

Round 1

Reviewer 1 Report

The Authors present the result of a survey suggesting that comorbidity of tobacco abuse in young adults suffering ADHD condition is likely to suggest other addictive behaviors. The result is interesting even if it is not new. The weakest point in the manuscript is the fact that the Authors have merely performed a statistical study on the survey of the Adult ADHD Self-Report Scale (ASRS-v1.1).  This questionnaire alone is insufficient to assess the degree of ADHD. Additional questionnaires, such as CAARS-SV, have provided much better results.  Moreover, the Authors write about adolescents anging 18-22 years old, but these individuals fall in the category of young adults and not adolescents.
   The ASRS questionnaire is rather used as a support for the psychiatrist to quantify some metrics of hyperactivity, but an accurate diagnosis can only be done through a clinical evaluation. Control subjects, without a diagnose of ADHD even after a clinical evaluation, may also present an ASRS score similar to clinically diagnosed patients. The CAARS-SV allows to determine the type of ADHD with a quantitative estimation of Inattentive symptoms, Hyperactive/impulsive symptoms and total ADHD index.
   The paper is also lacking important citations, showing that cholinergic system activity may be important in the cognitive deficits of ADHD and may be a useful therapeutic target (https://pubmed.ncbi.nlm.nih.gov/18022679/), showing that withdrawal from addictive compounds alters the functioning of the mesolimbic system (https://pubmed.ncbi.nlm.nih.gov/21886590/) and  showing that ADHD increased odds of e-cigarette use initiation but did not alter the shape of use trajectory among initiators (https://pubmed.ncbi.nlm.nih.gov/29304219/).
   The paper should be considerably reduced because of inherent weaknesses in the design of the questionnaires and the lack of clinical evaluation. The Introduction should be reduced accordingly. In the Discussion the Authors should clearly emphasize the limitation of their study in view of those weaknesses and open the discussion along the axis of the citations mentioned above.
   In conclusion, the paper should undergo a major revision or a rejection in its present form. 

Author Response

Thanks for your observation, we have modified the paper following your advices (where it possible).

We have not used the CAARS- SV because for our aim was better a screening test like ASRS (suggested by World Health Organizzation). But future study will include CAARS.

Reviewer 2 Report

The study investigates ADHD symptoms screened by the ASRS and nicotine, THC, alcohol and benzodiazepine use as well as gambling behaviour in a young adult population. The authors found relatively high prevalence of ADHD-positive screenings (15% of the sample) and those with increased ADHD symptoms also more frequently smoked cigarettes but none of the other abuse behaviours occured significantly more often in ADHD young adults. The study is of interest, however, I have several minor suggestions that should be addressed. 

  1. The authors term their study population "adolescents", however the age range was 18-22 years, so I would rather call them young adults. Or explain that you used the WHO definition of adolescence which is 10-20 years, but the ones older than that are still termed young adults.
  2. Why did the authors not use existing and validated questionnaires for substance use like for example the Fagerström for nicotine dependance, the AUDIT for alcohol abuse screening and the NIDA for other illegale substance abuse? And they should show an English translation of their own constructed questionnaire in the supplementary material. 
  3. What about e-cigarettes/nicotine inhalers? 
  4. Did the authors specifically ask for THC? Or Cannbis? Why not Cannabinoids and also CBD?
  5. What kind of gambling behaviour did they ask for? Also video games and online gambling? 
  6. Results: what about the alcohol use in the positive vs. the negative screened young adults? It is not mentioned in the results and not displayed in the table. 
  7. What does foreign father mean? What about foreign mothers?
  8. Why were so many more males included in the study? 
  9. Page 12, line 199: is not 
  10. Page 13, line 253: in the present study there was no hints for a more common BZD use in the participants with increased ADHD symptoms, so the conclusions cannot be made. And only the benzodiazepines and z-substances have an abuse and dependency risk, the other hypnotics do not have that. So if the authors cite a study that ADHD children are taking hypnotics they should go much more into detail regarding the previous findings, because that does not mean that those hypnotics  are benzos. 
  11. The ADHD prevalence rate of 15% is very high, the authors should discuss this and are the previous Italian data in young adults to compare? 
  12. Are there any other studies with regars to ADHD and substance abuse in general not clinical population that had similar findings? In clinical poulation for sure the substance use rates in ADHD patients are much higher especially with regards to cannbis and alcohol use, but this is a general population based study and a rather small sample so that also can be the reason for the negative findings besides cigarette smoking.  
  13. Conclusion: the conclusion cannot be made from the study results because the authors could only show an association with cigarette smoking not with other substance use or gambling. 

Author Response

We have modified the paper following your advices:

1-"young adults" is more correct than "adolescents" fo our study

2-We don't use AUDIT and Fagerstrom to reduce the time to complete the protocol. Moreover we have not used Audit and Fagerstrom because the aim is not measure the level of addiction

3-we do not investigate e-cig and inhalers

4-we use "THC" because we are interesting to illicit use of cannabis

5-the question is not specific, we do not ask which type of gambling. Video game is not included in gambling problems, they are included in internet addiction problems.

6-no significant result are reported

7-We do not ask about foreign mother. Foreign father could be condition young adults development, especially in a patriarchal family

8-It depend on class room composition

9-ok

10- we have modified the paper following your suggestions where it possible. In our study we have considered benzodiazepine only. 

11-It's very high, but it is a screening questionnaire, probably the diagnostic data could present a minor percentage

12- Thanks for your observation, I underline in the "limitation of the study" that the sample size is small. Future study will include bigger sample size.

13- We have modified the conclusion

Round 2

Reviewer 1 Report

The revised version has considered some comments, but I notice that the most critical point has not been addressed adequately. The Authors have indeed recognized that the participants to their study could not be diagnosed only on the basis of the ASRS questionnaire. However, the paper is strongly biased in that direction.
The present title of the paper cannot be accepted. The Authors should definitely withdraw ADHD from the title and only refer to the ASRS questionnaire. The Abstract too, is too much biased and should clearly state that the participants could not be confounded with ADHD, irrespective of a ASRS score close to the score attained by ADHD patients.
It is absolutely fundamental that any questionnaire (and ASRS is not necessarily the best, but one among several others) cannot be misinterpreted with a precise psychiatric diagnose. 

Author Response

Dear reviewer, thanks for your observation. I attach the tex revised following your advice.

We have modified the title and underline again that we talk about ADHD symptomatology (not diagnosis).